# Modified Primary Anastomosis Using an Intestinal Internal Drainage Tube for Crohn’s Disease: A Pilot Study

**DOI:** 10.3390/jcm12010364

**Published:** 2023-01-03

**Authors:** Aojian Deng, Shaopeng Zheng, Lianwen Yuan, Kaimin Xiang, Fen Wang

**Affiliations:** 1Department of Gastroenterology, The Third Xiangya Hospital, Central South University, Changsha 410013, China; 2Hunan Key Laboratory of Non-Resolving Inflammation and Cancer, Changsha 410013, China; 3Department of Gastroenterology, Zhongshan Hospital, Xiamen University, Xiamen 361005, China; 4Department of Geriatric Surgery, Second Xiangya Hospital, Central South University, Changsha 410011, China; 5Department of Gastroenterological Surgery, The Third Xiangya Hospital, Central South University, Changsha 410013, China

**Keywords:** Crohn’s disease, staged procedures, primary anastomosis

## Abstract

Purpose: Surgical treatment is an important part of the management of Crohn’s disease (CD). However, the current recommended staged procedures require two operations, with long hospital stays and high costs, while traditional primary anastomosis has a high risk of complications. Therefore, the aim of this study was to compare the clinical efficacy and safety of modified primary anastomosis using intestinal internal drainage tubes for CD. Methods: In this study, emergency and nonemergency CD patients were included separately. Then, the patients were divided into three subgroups: patients with intestinal internal drainage tubes (modified primary anastomosis), staged procedures, and traditional primary anastomosis. The main outcomes were the number of hospitalizations, length and cost of the first hospital stay, length and cost of total hospital stays, and complications. Results: The outcomes of the three subgroups of emergency CD patients were not significantly different. For nonemergency CD patients, patients with intestinal internal drainage tubes had shorter total hospital stays and fewer hospitalizations compared with the staged procedures subgroup, while no significant differences in any outcomes were observed between the modified and traditional primary anastomosis subgroups. Conclusions: For emergency CD patients, no significant advantage in terms of the main outcomes was observed for modified primary anastomosis. For nonemergency CD patients, modified primary anastomosis reduced the length of total hospital stays and hospitalizations compared with staged procedures. The placement of intestinal internal drainage tubes allows some patients who cannot undergo primary anastomosis to undergo it, which is a modification of traditional primary anastomosis.

## 1. Introduction

Crohn’s disease (CD) is an inflammatory bowel disease (IBD) of unknown etiology. The incidence of CD differs worldwide, with a high incidence in more developed countries and an increasing incidence in developing countries, imposing a heavy economic burden worldwide [1]. Most patients with CD exhibit a chronic inflammatory behavior at diagnosis, but many of them present with intestinal complications, including strictures, fistulas, and abscesses over time [2]. Despite major advances in the medical management of CD in recent decades, surgical treatment remains a critical part of CD treatment [3]. A total of 16.3% of patients require surgery at 1 year, 33.3% require surgery at 5 years, and approximately 50% require surgery at 10 years after diagnosis [4,5].

Current guidelines and studies recommend staged procedures in the case of a non-drainable abscess or steroid therapy with a dosage of >20 mg of prednisone, with bowel resection and protective stoma performed in the first stage and bowel anastomosis performed in the second stage. The protective stoma procedure redirects the contents of the bowel, which reduces the risk of postoperative intra-abdominal septic complications and is beneficial for alleviating the diseases [6,7,8,9,10]. Despite the abovementioned advantages of staged procedures, many limitations exist, such as the heavy damage by the second procedure, stoma-related complications, and expenditures, which can substantially reduce the quality of life of the patients and impose financial strain on their families [11]. For traditional primary anastomosis, the diseased bowel segment is resected with a primary anastomosis, which restores intestinal continuity in one stage and reduces complications and expenditures [12]. However, many studies have shown that traditional primary anastomosis increases the risk of anastomosis leakage, which is associated with a decreased quality of life for most patients. Hence, methods to reduce the burden and improve quality of life of CD patients without increasing the risk of traditional primary anastomosis complications has become a common concern for both doctors and patients.

In this study, after resecting the diseased bowel segment and performing primary anastomosis, an intestinal internal drainage tube was placed on the proximal side of the anastomosis through enteroscopy or direct insertion. By combining endoscopy and surgery, we hope to reduce the tension in the intestine around the anastomosis and the risk of anastomosis leakage, creating favorable conditions and improving the success rate of primary anastomosis, which avoids second-stage surgery. This study provides a reference for the surgical treatment of CD.

## 2. Materials and Methods

### 2.1. Study Objectives and Design

The study was conducted in accordance with the Declaration of Helsinki, and the protocol was approved by the Ethics Committee of Central South University (approval number I22177).

The inclusion criteria were as follows: from 2014 to 2021, patients who underwent surgery for CD or its complications at the Third Xiangya Hospital of Central South University and the Second Xiangya Hospital of Central South University.

The exclusion criteria for the subjects were a diagnosis of serious diseases in other systems, malignant tumors, and a serious lack of medical records.

The sex, ages at the diagnosis of CD and operation, interval between the diagnosis and operation, smoking history, previous operation history, preoperative Montreal classification, preoperative treatment, preoperative hemoglobin level, and white blood cell count of patients were collected as the preoperative baseline data.

Due to the substantial difference in the preoperative preparation (i.e., the severity of illness) between patients undergoing emergency and nonemergency surgery, we divided patients into an emergency group and a nonemergency group. Afterward, the patients in each group were divided into three subgroups according to the operative methods: (1) patients with an intestinal internal drainage tube (modified primary anastomosis), (2) staged procedures, and (3) traditional primary anastomosis. The number of hospitalizations, length and cost of the first hospital stay, length and cost of total hospital stays, and short-term postoperative complications were compared among the three subgroups in each group (the specific definition of each indicator is provided in Section 2.3).

### 2.2. Surgical Technique

In this study, we performed a modified primary anastomosis for CD. After resecting the diseased segment and performing primary anastomosis, a disposable medical drainage tube was used as the intestinal internal drainage tube and placed on the proximal side of the anastomosis by an enteroscope or direct insertion. The proximal side of the intestinal internal drainage tube was placed about 20 cm from the proximal side of the anastomosis, and the distal side of the intestinal internal drainage tube was fixed to the anal skin (Figure 1). The specific position is not fixed, as long as the tube can be guaranteed to cross the anastomosis, and 20 cm is the adequate position. The tension in the intestinal lumen at the anastomosis was reduced according to Pascal’s principle, providing a good condition for primary anastomosis. After surgery, patients received postoperative nutritional support and the intestinal internal drainage tube was directly removed from the anal canal, eliminating the need for the second stage.

### 2.3. Data Collection

All baseline data for patients were collected before surgery, unless specified otherwise. The Montreal classification was based on endoscopy reports, imaging, surgery, and clinical notes. Perianal disease was defined as a previous anal abscess, fistula, stricture, or surgery for anal disease. Smokers were patients who continued to smoke until surgery. Previous surgery was defined as patients who had any other surgical experience before this surgery. Preoperative treatment was defined as the treatment associated with CD, including supportive care, mesalazine, immunosuppressants, and biological agents. Patients receiving supportive care were newly diagnosed and had not received specific treatment. Immunosuppressants mainly included azathioprine, and biological agents included vedolizumab and infliximab. Patients receiving corticosteroids just before the surgery were classified as hormones.

The number of hospitalizations was defined as the number of hospitalizations that were directly related to this surgery (e.g., stoma retraction, surgery-related complications, etc.). The length of first hospital stay was the interval between surgery and discharge reported in days. The cost of the first hospital stay was the total expense of the current hospitalization due to the surgery. The length of the total hospital stay was the number of days of hospital stays directly related to this surgery. The cost of total hospital stays was total expense of hospitalizations directly related to this surgery. Short-term postoperative complications were defined as an incision infection, anastomotic leak, intra-abdominal septic complications (IASC), etc.

### 2.4. Statistical Analysis

Because of the substantial differences in baseline characteristics in nonemergency patients, we used propensity score matching (PSM) to separately match patients who underwent modified primary anastomosis with patients who underwent staged procedures and those who received traditional primary anastomosis and to minimize the effects of other confounders on the outcomes. Patients who underwent modified primary anastomosis were separately matched in a 1:1 ratio with patients treated with staged procedures and those who underwent traditional primary anastomosis with a 0.1 match tolerance based on the Montreal classification and preoperative treatment.

IBM SPSS Statistics 26 was used for statistical analyses. Categorical variables were reported as frequencies or percentages and were compared using chi-square or Fisher’s exact tests, as required by the test method. Categorical variables are presented as frequencies or percentages, and the chi-square test or Fisher’s exact test was used to perform the comparison according to the requirements of the test methods. Quantitative variables are presented as medians and interquartile ranges (IQRs), and the nonparametric Kruskal–Wallis test and the Mann–Whitney rank sum test were used to determine statistical significance.

## 3. Results

### 3.1. Emergency Group

#### 3.1.1. Baseline Characteristics

Based on the inclusion and exclusion criteria, 112 patients were enrolled. Among these patients, 17 received emergency surgery, including 5 modified primary anastomosis procedures, 2 staged procedures, and 10 traditional primary anastomosis procedures. The baseline profiles of the emergency patients are listed in Table 1. None of the patients had a family history of CD or a special family history.

#### 3.1.2. Clinical Outcomes

We compared the clinical outcomes between patients receiving modified primary anastomosis, staged procedures, and traditional primary anastomosis in emergency situations. Modified primary anastomosis did not show significant superiority in terms of the number of hospitalizations, length and cost of the first hospital stay, length and cost of total hospital stays, or short-term postoperative complications (*p* > 0.05). The results are summarized in Table 2 and Figure 2.

### 3.2. Nonemergency Group

#### 3.2.1. Baseline Characteristics

Based on the inclusion and exclusion criteria, 95 nonemergency patients were enrolled in this study, including 16 who underwent modified primary anastomosis, 34 treated with staged procedures, and 45 who received traditional primary anastomosis. Since large differences in baseline characteristics were observed, we conducted PSM to control for these differences. After matching, 16 patients who underwent modified primary anastomosis, 15 who underwent staged procedures, and 15 who underwent traditional primary anastomosis were included in the final analysis. No significant differences in the baseline characteristics between were observed the modified primary anastomosis group and the other two control groups after matching (Table 3). None of the patients had a family history of CD or a special family history.

#### 3.2.2. Clinical Outcomes

The clinical outcomes of nonemergency patients obtained after matching are presented in Table 4 and Figure 3. The results of the comparison of modified primary anastomosis and staged procedures groups indicated that the length of total hospital stay was significantly shorter in patients undergoing modified primary anastomosis patients (*p* = 0.045), and the number of hospitalizations was also reduced (*p* = 0.009). No significant differences were observed between patients undergoing modified primary anastomosis and those undergoing traditional primary anastomosis (*p* > 0.05). Significant differences in short-term postoperative complications were not observed among the three groups (*p* > 0.05).

## 4. Discussion

In this study, after resection of the segment and primary anastomosis, we placed the intestinal internal drainage tube about 20 cm from the proximal side of the anastomosis through enteroscopy or direct insertion. Pascal’s principle was used to directly transmit the pressure in the bowel at the anastomosis to the anus, normalizing the pressure from the anastomotic site to the anus. Therefore, the tension of intestinal fluid and gas around the anastomosis on the intestinal wall are relieved such that the anastomosis is in a clean and low-tension state, providing a better physiological environment for the recovery of the anastomosis and achieving better primary anastomosis.

Prior to surgery, the surgery option was determined based on the preoperative condition of CD patients, which also exerted a substantial effect on the clinical outcomes. Patients with poor preoperative conditions were more likely to choose staged procedures. On the one hand, CD and its complications affect the absorption of nutrients, resulting in poor nutritional outcomes, which increase the risk of postoperative complications such as anastomotic leakage [13,14,15]. On the other hand, most CD patients are treated with glucocorticoids and immunosuppressants, which also increases the risk of anastomotic leakage and other postoperative complications [16,17]. Heterogeneity existed between the studies regarding the effects of biological agents on postoperative outcomes [18,19]. We found that the preoperative hemoglobin level in patients treated with modified primary anastomosis was significantly lower than that of patients treated with traditional primary anastomosis. Thus, modified primary anastomosis may allow patients who would otherwise need to undergo staged procedures due to severe disease to undergo primary anastomosis, avoiding the complications of staged procedures and expanding indications for traditional primary anastomosis.

Many problems are associated with the existing staged procedures and traditional primary anastomosis for CD. The first is the large burden of surgery. Approximately 5% of patients require reoperation due to disease recurrence within 1 year after surgery, imposing an immense burden on the health care infrastructure and patients [9,20]. For staged procedures, problems with the stoma also increase the burden on patients and hospitals [21,22]. The second is that the incidence of postoperative complications is as high as 21% [23]. Many complications caused by stoma have been noted in staged procedures, including stoma-related complications, water electrolyte disorder, and acidolysis disorder [24,25,26]. Traditional primary anastomosis may increase the risk of anastomosis leakage and IASC [27,28]. Hence, we compared the length and cost of the hospital stay and short-term postoperative complications of modified primary anastomosis with the existing staged procedures and traditional primary anastomosis.

For emergency patients, no significant differences in clinical outcomes were observed between modified primary anastomosis, the staged procedures, and traditional primary anastomosis. The potential explanations for these results are described below. First, the number of emergency patients was too small to draw meaningful conclusions. Second, although little difference was observed in the preoperative baseline characteristics of the enrolled patients, the comparison did not reflect whether differences in the preoperative conditions of the patients existed. Due to the urgency of the emergency patients, the heterogeneity of the preoperative conditions of different patients was large, such as the time from admission to operation and nutritional preparation. Third, the technical proficiency and level of the surgeon exert substantial effects on the clinical outcomes of emergency surgery, which are not adequately reflected in the study. Hence, we were unable to determine the superior and inferior results of modified primary anastomosis, the staged procedures, and traditional primary anastomosis in this study.

For nonemergency CD patients, modified primary anastomosis substantially reduced the length of the total hospital stay and number of hospitalizations compared with staged procedures, without increasing the cost of hospital stay and short-term postoperative complications. No significant differences in clinical outcomes were observed between the modified and traditional primary anastomosis groups.

We identified advantages for the use of intestinal internal drainage tubes over staged procedures potentially because patients undergoing staged procedures had a longer disease duration and longer-term treatment with hormones and immunosuppressants, leading to poor outcomes. However, patients who received modified primary anastomosis often had a shorter disease duration and received less clinical treatment with hormones and immunosuppressants. When PSM was performed, we were unable to match patients undergoing staged procedures with more severe conditions (such as complete intestinal obstruction and poor nutritional status), which may create a bias. Hence, for patients with a shorter disease duration and less treatment with hormones or immunosuppressants, modified primary anastomosis might reduce the length of total hospital stay and number of hospitalizations without increasing the cost of the hospital stay and short-term postoperative complications, while further studies are needed for patients with a longer disease duration and more long-term treatment with hormones or immunosuppressants.

Due to the insufficient number of nonemergency patients, this study was unable to show a significant difference between modified and traditional primary anastomosis. However, as outlined above, we found that modified primary anastomosis may allow patients who would otherwise need to undergo staged procedures to undergo primary anastomosis, expanding indications for traditional primary anastomosis. Modified primary anastomosis also did not increase the length or cost of the hospital stay or the risk of short-term postoperative complications. Hence, modified primary anastomosis may serve as an alternative procedure for CD patients, which requires further investigation.

The limitations of this study are described below. First, as a pilot study of a modified procedure, the number of patients included in this study was small, allowing us to draw only limited conclusions. Second, studies on patients with more severe conditions were lacking, and some preoperative baseline data were incomplete, such as data on Crohn’s disease activity index (CDAI), fecal calprotectin levels, preoperative nutrition, and other baseline indicators of CD. Third, the follow-up duration in this study was insufficient, and a study examining long-term complications and control of CD was lacking. Fourth, the IQR of the internal drainage tube group in emergency patients and the IQR of the staged procedures group in nonemergency patients were markedly broad, which means that there is greatly heterogeneity among enrolled patients in these group. Further studies should be conducted to increase the number of patients enrolled, randomize the surgical option, stratify the severity of the patient’s condition, and extend the follow-up duration as approaches to explore the long-term outcomes of modified primary anastomosis.

## 5. Conclusions

For emergency CD patients, this study was unable to show a significant difference between modified primary anastomosis, the existing staged procedures, and traditional primary anastomosis. For nonemergency CD patients, compared with staged procedures, the modified primary anastomosis significantly reduced the length of the total hospital stay and number of hospitalizations without increasing the cost of hospital stays and the risk of short-term postoperative complications. This procedure may expand the indications for traditional primary anastomosis and serve as an alternative treatment for CD patients.

## Figures and Tables

**Figure 1 jcm-12-00364-f001:**
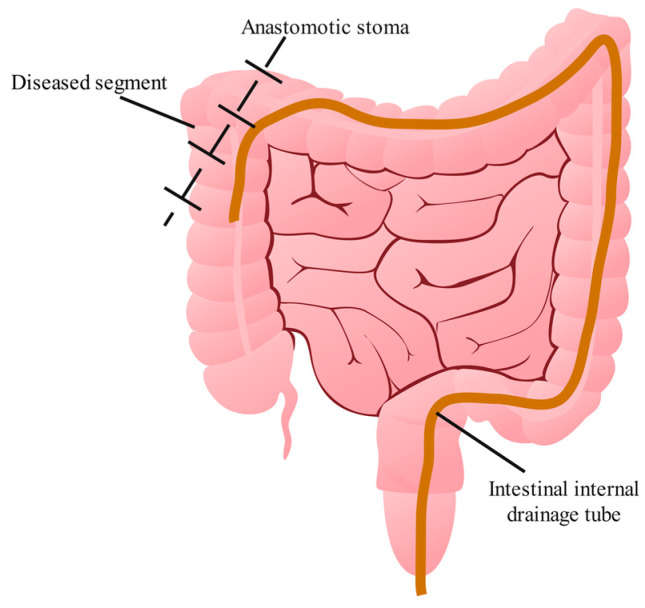
Schematic diagram of modified primary anastomosis.

**Figure 2 jcm-12-00364-f002:**
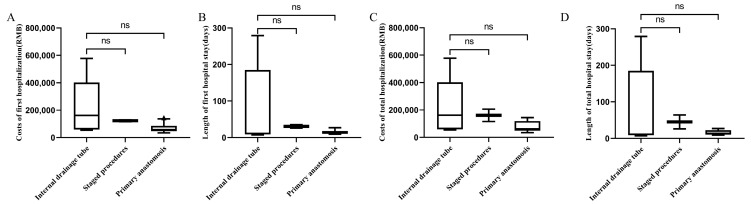
Lengths and costs of the first and total hospital stays for emergency patients. (**A**) Costs of the first hospitalization; (**B**) length of the first hospital stay; (**C**) costs of total hospitalization; (**D**) length of total hospitalization. ns: *p* > 0.05.

**Figure 3 jcm-12-00364-f003:**
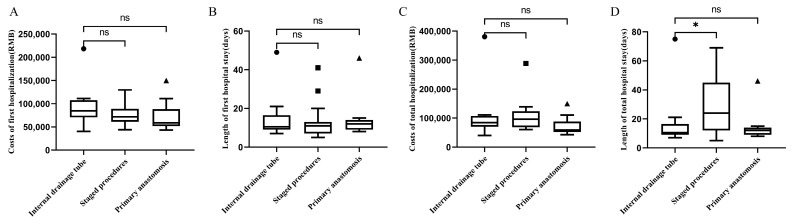
Lengths and costs of the first and total hospital stays for nonemergency patients. (**A**) Costs of the first hospitalization; (**B**) length of the first hospital stay; (**C**) costs of total hospitalization; (**D**) length of total hospitalization. ns: *p* > 0.05, * *p* < 0.05.

**Table 1 jcm-12-00364-t001:** Baseline characteristics of emergency patients.

Variables	Internal Drainage Tube (*n* = 5)	Staged Procedures (*n* = 2)	Primary Anastomosis (*n* = 10)	*p* Value
Male, *n* (%)	4 (80.0)	1 (50.0)	10 (100.0)	0.154
Age at diagnosis (IQR)	46.0 (42.0–54.5)	36.0 (18.0–54.0)	34.5 (30.8–47.8)	0.147
Age at surgery (IQR)	47.0 (45.5–57.0)	36.5 (18.3–54.8)	37.0 (33.3–53.8)	0.167
Duration of disease (IQR)	1.0 (0.5–5.5)	0.5 (0.3–0.8)	0.0 (0.0–4.3)	0.459
Smokers, *n* (%)	3 (60.0)	0 (0.0)	5 (50.0)	0.620
Previous surgery, *n* (%)	2 (40.0)	2 (100.0)	6 (60.0)	0.401
Montreal classification, *n* (%)				
A				0.047
A2, 17–40 years	0 (0.0)	1 (50.0)	6 (60.0)	
A3, >40 years	5 (100.0)	1 (50.0)	4 (40.0)	
L				1.000
L1, ileal	4 (80.0)	2 (100.0)	7 (70.0)	
L2, colonic	0 (0.0)	0 (0.0)	1 (10.0)	
L3, ileocolonic	1 (20.0)	0 (0.0)	2 (20.0)	
B				0.620
B2, structuring	3 (60.0)	0 (0.0)	5 (50.0)	
B3, penetrating	2 (40.0)	2 (100.0)	5 (50.0)	
Perianal disease (*p*)	0 (0.0)	0 (0.0)	3 (30.0)	0.669
Preoperative treatment, *n* (%)				0.482
Supportive care	2 (40.0)	1 (50.0)	7 (70.0)	
Mesalazine	1 (20.0)	0 (0.0)	2 (20.0)	
Immunosuppressant	1 (20.0)	1 (50.0)	1 (10.0)	
Biological agents	1 (20.0)	0 (0.0)	0 (0.0)	
Hormones, *n* (%)	0 (0.0)	1 (50.0)	0 (0.0)	0.118
Hemoglobin, g/L (IQR)	79.0 (65.5–112.0)	101.5 (50.8–152.3)	117.0 (72.5–146.3)	0.372
Leucocytes, ×109 cells/L (IQR)	6.6 (4.8–8.4)	8.4 (4.2–12.6)	9.4 (6.4–15.6)	0.336

**Table 2 jcm-12-00364-t002:** Clinical outcomes of emergency patients.

Variables	Internal Drainage Tube (*n* = 5)	Staged Procedures (*n* = 2)	Primary Anastomosis (*n* = 10)	*p* Value
Number of hospitalizations (%)				0.301
1	5 (100.0)	1 (50.0)	9 (90.0)	
2	0 (0.0)	1 (50.0)	1 (10.0)	
Length of the first hospital stay, days (IQR)	11.0 (8.5–185.0)	30.5 (15.3–45.8)	14.0 (10.0–18.0)	0.311
Costs of the first hospitalization, RMB (IQR)	161,801.1 (59,100.7–402,191.5)	122,680.7 (61,340.4–184,021.1)	58,033.8 (46,993.5–85,779.8)	0.098
Length of the total hospital stay, days (IQR)	11.0 (8.5–185.0)	45.0 (22.5–67.5)	14.0 (10.0–22.5)	0.389
Costs of total hospitalization, RMB (IQR)	161,801.1 (59,100.7–402,191.5)	160,778.3 (80,389.2–241,167.5)	61,413.4 (48,904.5–119,058.5)	0.096
Any complications, *n* (%)	2 (40.0)	2 (100.0)	3 (30.0)	0.294

IQR, interquartile range.

**Table 3 jcm-12-00364-t003:** Baseline characteristics of nonemergency patients before and after PSM.

Variables	Before Matching			*p* Value ^#^	*p* Value ^##^	After Matching			*p* Value ^#^	*p* Value ^##^
	Internal Drainage Tube (*n* = 16)	Staged Procedures (*n* = 34)	Primary Anastomosis (*n* = 45)			Internal Drainage Tube (*n* = 16)	Staged Procedures (*n* = 15)	Primary Anastomosis (*n* = 15)		
Male, *n* (%)	9 (56.3)	25 (73.5)	38 (84.4)	0.330	0.036	9 (56.3)	12 (80.00)	12 (80.00)	0.252	0.252
Age at diagnosis (IQR)	32.5 (27.0–49.3)	29.0 (24.5–41.3)	33.0 (27.0–48.5)	0.422	0.837	32.5 (27.0–49.3)	29.0 (23.0–42.0)	40.0 (30.0–57.0)	0.495	0.101
Age at surgery (IQR)	33.0 (28.3–53.8)	34.0 (29.5–45.3)	35.0 (29.0–50.0)	0.884	0.730	33.0 (28.3–53.8)	36.0 (27.0–47.0)	41.00 (30.0–61.0)	0.800	0.072
Duration of disease (IQR)	0.0 (0.0–1.8)	2.5 (0.8–6.0)	0.0 (0.0–1.5)	0.007	0.789	0.0 (0.0–1.8)	3.0 (0.0–6.0)	0.0 (0.0–1.0)	0.060	0.953
Smokers, *n* (%)	4 (25.0)	5 (14.7)	21 (46.7)	0.442	0.152	4 (25.0)	3 (20.0)	9 (60.0)	1.000	0.073
Previous surgery, *n* (%)	6 (37.5)	23 (67.6)	23 (51.1)	0.066	0.395	6 (37.5)	7 (46.7)	10 (66.7)	0.722	0.156
Montreal classification, *n* (%)										
A				1.000	0.766				1.000	0.156
A2, 17–40 years	11 (68.8)	22 (64.7)	28 (62.2)			11 (68.8)	10 (66.7)	6 (40.0)		
A3, >40 years	5 (31.3)	12 (35.3)	17 (37.8)			5 (31.3)	5 (33.3)	9 (60.0)		
L				0.762	0.381				0.768	0.088
L1, ileal	6 (37.5)	14 (41.2)	23 (51.1)			6 (37.5)	4 (26.7)	10 (66.7)		
L2, colonic	1 (6.3)	5 (14.7)	6 (13.3)			1 (6.3)	2 (13.3)	2 (13.3)		
L3, ileocolonic	9 (56.3)	15 (44.1)	16 (35.6)			9 (56.3)	9 (60.0)	3 (20.0)		
B				0.406	0.262				0.333	1.000
B2, structuring	15 (93.8)	28 (82.4)	45 (100.0)			15 (93.8)	12 (80.0)	15 (100.0)		
B3, penetrating	1 (6.3)	6 (17.6)	0 (0.0)			1 (6.3)	3 (20.0)	0 (0.0)		
Perianal disease (*p*)	4 (25.0)	14 (41.2)	12 (26.7)	0.351	1.000	4 (25.0)	4 (26.7)	5 (33.3)	1.000	0.704
Preoperative treatment, *n* (%)				0.024	0.253				0.226	1.000
Supportive care	12 (75.0)	10 (29.4) *	25 (55.6)			12 (75.0)	6 (40.0)	11 (73.3)		
Mesalazine	2 (12.5)	9 (26.5)	7 (15.6)			2 (12.5)	5 (33.3)	2 (13.3)		
Immunosuppressants	1 (6.3)	11 (32.4) *	12 (26.7)			1 (6.3)	3 (20.0)	2 (13.3)		
Biological agents	1 (6.3)	4 (11.8)	1 (2.2)			1 (6.3)	1 (6.7)	0 (0.0)		
Hormones, *n* (%)	1 (6.3)	1 (2.9)	3 (6.7)	0.542	1.000	1 (6.3)	1 (6.7)	1 (6.7)	1.000	1.000
No. of hemoglobin, *n* (%)	16 (100.0)	32 (94.1)	39 (86.7)			16 (100.0)	14 (93.3)	15 (100.0)		
Hemoglobin, g/L (IQR)	106.0 (97.3–127.3)	111.5 (93.5–122.8)	124.0 (114.0–137.0)	0.632	0.024 *	106.0 (97.3–127.3)	114.0 (69.8–119.0)	123.0 (104.0–134.0)	0.470	0.299
No. of leucocytes, *n* (%)	16 (100.0)	32 (94.1)	39 (86.7)			16 (100.0)	14 (93.3)	15 (100.0		
Leucocytes, ×109 cells/L (IQR)	6.5(4.6–7.5)	5.3 (4.3–8.1)	6.1 (4.3–8.1)	0.670	0.993	6.5(4.6–7.5)	6.1 (4.3–9.6)	6.7 (4.3–8.6)	0.822	0.545

IQR, interquartile range; ^#^ *p* value for the comparison between the internal drainage tube and staged procedures; ^##^ *p* value for the comparison between the internal drainage tube and primary anastomosis. * *p* < 0.05.

**Table 4 jcm-12-00364-t004:** Clinical outcomes of nonemergency patients after PSM.

Variables	Internal Drainage Tube (*n* = 16)	Staged Procedures (*n* = 15)	Primary Anastomosis (*n* = 15)	*p* Value ^#^	*p* Value ^##^
Number of hospitalizations (%)				0.009 **	0.484
1	14 (87.5)	6 (40.0)	15 (100.0)		
2	2 (12.5)	9 (60.0)	0 (0.0)		
Length of the first hospital stay, days (IQR)	10.5 (9.0–16.5)	11.0 (7.0–13.0)	12.0 (9.0–14.0)	0.861	0.922
Costs of the first hospitalization, RMB (IQR)	84,412.0 (70,513.5–107,635.0)	71,548.4 (60,887.7–88,917.0)	58,684.2 (51,674.8–88,149.7)	0.093	0.129
Length of the total hospital stay, days (IQR)	10.5 (9.0–16.5)	24.0 (12.0–45.0)	12.0 (9.0–14.0)	0.045 *	0.922
Costs of total hospitalization, RMB (IQR)	84,412.0 (70,513.5–107,635.0)	96,154.7 (68,891.4–123,917.0)	58,684.2 (51,674.8–88,149.7)	0.318	0.129
Any complication, *n* (%)	2 (12.5)	4 (26.7)	2(13.3)	0.394	1.000

IQR, interquartile range; ^#^ *p* value for the comparison between the internal drainage tube and staged procedures; ^##^ *p* value for the comparison between the internal drainage tube and primary anastomosis. * *p* < 0.05, ** *p* < 0.01.

## Data Availability

The data presented in this study are available on request from the corresponding author.

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
