# Peer review of "Modified Primary Anastomosis Using an Intestinal Internal Drainage Tube for Crohn’s Disease: A Pilot Study"

_jcm, 2023, doi:10.3390/jcm12010364_

Round 1

Reviewer 1 Report

In this manuscript, Dr. Deng et al. report an intestinal drainage tube's clinical efficacy and safety for Crohn's disease. Although there was no advantage for the emergency patients, the authors showed that the length of total hospital stays and hospitalizations were reduced for the nonemergency patients. This report could provide new insight into the surgical treatments for Crohn's disease. However, more detailed methods and discussion would be required for this paper. 

Specific recommendations for revision-a) major

Line 101: Though authors described that the distal side of the tube was fixed to the anal skin, how the proximal side was confirmed to be placed at the adequate position, 20 cm from the proximal side of the anastomosis? Readers would wonder if the position could change during the follow-up. Did the physician of the participants check it with an X-ray regularly?

Line 116: As the authors mentioned in the discussion, corticosteroid use increases the risk of postoperative complications. Detailed preoperative treatments were required; however, readers needed to know the proportions of corticosteroid use in the participants. Also, "supportive care," "immunosuppressant," and "hormones" was not defined in the manuscript.  

Figures showed that the IQR of the internal drainage tube group in emergency patients and the IQR of the staged procedures group in nonemergency patients were markedly broad. The discussion about this fact may provide insights into using this unique method in clinical situations. 

Specific recommendations for revision-b) minor:

Tables: The data could be rounded off to the first decimal place since the number in the second decimal place is unimportant in clinical settings.

Abstract: In conclusion, the last sentence must be revised since "undergo 0primary anastomosis" was duplicated. 

Tables: **P and *** PP were defined below tables, but they were not used in any Tables. On the other hand,  Table 3 included *, which was not defined. 

Reviewer 2 Report

-       -  “Current guidelines and studies recommend staged procedures, with bowel resection and protective stoma performed in the first stage and bowel anastomosis performed in the second stage.”

This is absolutely not true. The classic patient with stenosing Crohn's disease in terminal ileum is operated in a single stage.

On the other hand, in the case of a non-drainable abscess or steroid therapy with a dosage of >20 mg of prednisone, the resection with anastomosis is done in the first time, and in the second time the stoma is closed.

-        - “The tension in the intestinal lumen at the anastomosis was reduced”

How did you measure it?

-        - “during routine suture”

What suture?

-       - “None of the patients had a family history of CD or a special family history.”

Report this sentence in the results and not in the methods

-        - Increase the size of the figures

-        - Was the anastomotic technique the same throughout the period or did you change it? For example, have you introduced the Kono anastomosis in recent years?

-        - Use oxford comma

Round 2

Reviewer 1 Report

Thank you for revising the manuscript.

Reviewer 2 Report

Thank you for your corrections.